# Federated learning based futuristic biomedical big-data analysis and standardization

**Afifa Salsabil Fathima**[1], **Syed Muzamil Basha**[1], **Syed Thouheed Ahmed**[2], **Sandeep Kumar Mathivanan**[3], **Sukumar Rajendran**[4], **Saurav Mallik**[5,6]*, **Zhongming Zhao**[6,7]*

**1** School of Computer Science and Engineering, REVA University, Bengaluru, India, **2** Department of Electrical Engineering, Indian Institute of Technology, Hyderabad, India, **3** School of Computing Science and Engineering, Galgotias University, Greater Noida, India, **4** School of Computing Science and Engineering, VIT Bhopal University, Sehore, MP, India, **5** Department of Environmental Health, Harvard T H Chan School of Public Health, Boston, MA, United States of America, **6** Center for Precision Health, School of Biomedical Informatics, The University of Texas Health Science Center at Houston, Houston, TX, United States of America, **7** Human Genetics Center, School of Public Health, The University of Texas Health Science Center at Houston, Houston, TX, United States of America

* Zhongming.Zhao@uth.tmc.edu (ZZ); sauravmtech2@gmail.com (SM)

**Data Availability Statement:** Data will be available in github: https://github.com/syedthouheed128/EHR-Standardization-Datasets.

## Abstract

Medical data processing and analytics exert significant influence in furnishing dependable decision support for prospective biomedical applications. Given the sensitive nature of medical data, specialized techniques and frameworks tailored for application-centric processing are imperative. This article presents a conceptualization for the analysis and uniformitarian of datasets through the implementation of Federated Learning (FL). The realm of medical big data stems from diverse origins, necessitating the delineation of data provenance and attribute paradigms to facilitate feature extraction and dependency assessment. The architecture governing the data collection framework is intricately linked to remote data transmission, thereby engendering efficient customization oversight. The operational methodology unfolds across four strata: the data origin layer, data acquisition layer, data classification layer, and data optimization layer. Central to this endeavor are multi-objective optimal datasets (MooM), characterized by attribute-driven feature cartography and cluster categorization through the conduit of federated learning models. The orchestration of feature synchronization and parameter extraction transpires across multiple tiers of neural networking, culminating in the provisioning of a steadfast remedy through dataset standardization and labeling. The empirical findings reflect the efficacy of the proposed technique, boasting an impressive 97.34% accuracy rate in the disentanglement and clustering of telemedicine data, facilitated by the operational servers within the ambit of the federated model.

## I. Introduction

The biomedical data and relative sources for the creation of interdependent-data, termed as meta-data has increased exponentially. The biomedical data digitalization has improved the

**Funding:** The author(s) received no specific funding for this work.

**Competing interests:** The authors have declared that no competing interests exist.

quality of service (QoS) and the decision capabilities of medical treatment and handling approaches. The data internally, has an enormous information with a classified and categorized parameter. Hence the demand for optimizing heaps of biomedical data is a challenging task. With the technological enhancements, the improvised approaches such as analytics, provide a reliable support to understand, classify and categorize the data into various sub-heaps of information indexing databases. Typically, the tread of data-optimization and classifying is backup with storage servers and accessing terminologies. The approaches can be tracked backed to Storage Area Networks (SAN) in storing and providing a reliable backup for uploaded information. With enhancements, the SAN based systems were developed to Cloud based system via interconnected servers.

The interconnected server nodes within the cloud ecosystem have established a pseudo networking link, resulting in the accumulation of a substantial volume of unclassified data. This phenomenon can be characterized as a consequential outcome of extensive data computational systems. The strategy of centralizing data storage and processing has garnered substantial support from a plethora of technological tools and methodologies. A significant challenge that arises pertains to the regulation of data volume and the management of data indexing flow originating from the centralized servers intended for communication purposes. A prospective solution involves a shift towards decentralizing data storage and confining processing activities primarily to edge devices. This shift is underpinned by the forthcoming paradigm of comprehensive data categorization and classification on a large scale. Furthermore, the augmentation of machine learning-based processing accentuates the necessity to critically assess and validate data stabilization and training models, all the while circumventing the need for centralized data storage.

The terminology of Federated Learning (FL) is based on the ideal concept of "NOT to store the data" hence it follows the approach of de-centralizing the data storage and processing via the participating devices. These devices are edge or terminating devices which either generates the data or participates in training the data models for reliable decision support. Within the confines of this article, an innovative approach to the field of biomedical big-data analytics is brought to the forefront. The primary objective of this approach revolves around the establishment of a dependable and high-performance model or framework. This model is meticulously designed to address the intricate challenges associated with the optimization and standardization of vast and intricate biomedical datasets across a spectrum of heterogeneous networking devices and intricate infrastructural components.

The significance of this proposed approach lies in its potential to revolutionize the manner in which biomedical data is harnessed, processed, and ultimately utilized for informed decision-making. By meticulously optimizing and standardizing data, the proposed model seeks to enhance data coherence, consistency, and reliability, thereby facilitating more accurate and meaningful analyses. In the context of modern healthcare, where a deluge of complex and diverse data is generated from various sources, ranging from medical imaging equipment to wearable devices, the need for a robust approach to harmonize and streamline these datasets is paramount.

The objective of proposed model is to bridge the gap by offering a systematic and coherent solution that caters to the intricacies of different data sources, networking protocols, and infrastructure topologies for biomedical big-data analysis and processing. Through a meticulous integration of cutting-edge techniques from fields such as data science, machine learning, and network engineering, this approach endeavors to unlock the true potential of biomedical big data. By providing a solid foundation for data optimization and standardization, this model stands poised to propel the field of biomedical analytics into a new era of enhanced insights and discoveries. The model is bound to develop the federated learning based technique for

 

classifying and customizing the datasets. The article is organized with an introduction and literature survey in section I and II followed by a proposed model design in section III and mathematical proof of concept in section IV. The results and discussion with conclusion is discussed in section V and VI respectively.

## II. Literature survey

The Federated Learning (FL) is an improved version of technological development built from machine learning and advance networking tools. The concept of federated learning (FL) is to process and train the model without storing the data into a designated server. The approach is to develop a sustainable model via remote/edge devices connected to form a networking setup in training and customizing the overall model with upgraded information and training sets. The various challenges, approaches and ecosystem for federated learning is discussed and reported by [1] using heterogeneous and massive networking ecosystem. The concept of implementing a distributed learning via FL in heterogeneous network is complicated at the scenario of biomedical data processing. Since the biomedical data is sensitive and classified under the mode of storing and accessing. Hence a defined and standard operational protocol is needed. The process is supported by customizing the Electronic Health Records (EHR). [2] discussed the provision for designing and developing EHR based on federated learning approaches. This approach is based on prediction of models under heterogeneous networks.

Since the data sharing and training the model by a contributive approach in federated learning models is based on edge devices. These devices are bound to operate in given security conditions, hence the security of FL based networking model is a concern. [3] has discussed various aspects and reality of security concerns under data privacy. The researchers have developed and validated these techniques [4–6] in FL ecosystem development and processing the data under a secure and reliable manner in a distributed ecosystem. The FL applications [7] are wider and have a larger scope of operation under the given technological limitations. The biomedical data processing is one of the most influential paradigm of FL based model learning as discussed in [8]. The technique discusses the scope and future of digital health under Federated Learning ecosystem to support the fundamental arguments of information transmission and processing via remote/edge devices.

The dataset used in this experimental setup is Multi-Objective Optimal Medical (MooM) datasets [9] the dataset are processed and streamed into a standardized manner for faster, reliable and secure transmission under the networking ecosystem. The MooM datasets are claimed by a dedicated transferring protocol namely TelMED [10] to assure the data under dynamic user classification and clustering. The TelMED protocol is a reliable technique for distributed computation [10, 11]. These approaches internally support and validate the purpose of building a reliable and self-learning ecosystem for decision making. In this article, the proposed framework assures the reliable and sustainable model design using federated learning towards optimizing the datasets and channel data. The recommendation support is extended from a learning approach of big-data collected from application front [12, 13]. These datasets acquire a larger portion of server space as it is independent of medical data standards.

In recent times, many efforts are in progress to assure and propose medical standards for data transmission and representation on a server. The approach [14] of personalization based data generation in EHR and recommendation has improved the analysis and standardization. The approach assures the data is categorized into recommended categories and further segmented on server (proposed storage space). Further the standardization is discussed on fair data storage and utilization [15] with respect to biomedical datasets. [15] discusses on the reliability fragments of data storages and accessing. A dedicated standard recommendation [16]

termed as Biomedical and Health Informatics (BHMI) is proposed by The International Medical Informatics Association (IMIA) for educating the biomedical data standardization across multiple platforms.

In general, the existing studies are dependent on the base foundation values such as architecture of storage system, configuration of servers, alignment of interdependency topologies and administrative privileges. These components provide an unrealistic approach for biomedical data transfer and standardization. Hence the proposed approach is developed with an objective to assure a reliable data standardization and analysis platform for biomedical big-datasets.

## III. System design

The federated learning (FL) modes are based on remote/edge node training and contributive learning from the connected and inter-connected devices. The data fragmentation and supporting approaches are bound to perform a regional approach of data categorization and classifying under a given time internal. The proposed diagram is categorized into four layers as shown in Fig 1. The primary and first layer is the 'Data Origin Layer (DOL)', it is responsible for data collection and processing via various edge/remote devices connected or participated in the processing of computational process. The devices data is categorized into the internal sources such as origin, instrumental or interdependent via multiple hopping. Each of data is loaded into the repository via an ordered arrangement of warehouse and dataset collections. This provides the ecosystem a stability to read, analysis, validate and record the information tracks and origin points. In the first layer, the overall contribution of data collection and processing is designed and developed.

The architectural composition advances with the incorporation of the 'Data Acquisition Layer (DAL)' as the second tier, closely followed by the 'Data Classification Layer (DCL)' in the third tier. These two layers synergistically intertwine, yielding a cohesive and synchronized framework essential for orchestrating the coordination and processing of data. Their cardinal objective is to effectuate the metamorphosis of raw, unstructured data residing within the repository into an organized and structured format. Within the realm of the 'Data Acquisition Layer (DAL),' the primary thrust lies in the utilization of elementary yet indispensable machine learning techniques, such as data filtering and alignment. These techniques play a pivotal role in imposing a coherent structure upon the incoming data influx. The data, post this preliminary transformation, is rendered amenable to subsequent layers, characterized by enhanced consistency and applicability.

Progressing to the subsequent stratum, the 'Data Classification Layer (DCL)' comes to the fore. Its raison d'être revolves around the classification and categorization of the preprocessed data into discrete classes or categories. This classification serves as a precursor to more intricate analyses and processing steps, as it empowers subsequent layers with a refined and intelligible input. The culmination of this sequential progression manifests in the 'Data Optimization Layer (DPL)' positioned as the fourth tier. This layer is dedicated to a multifaceted role involving intricate data manipulation and selective extraction. It operates with precision to transform the preprocessed data streams into a structured assemblage of distinct clusters. Each of these clusters, serving as an encapsulation of data points, embodies a coherent set of features, thereby facilitating a seamless avenue for data standardization and optimization. To gain a profound visual insight into this intricate stratified architecture, Figs 2 and 3 is presented as a definitive elucidation. It visually underscores the hierarchical alignment of each layer in precise consonance with the prescribed operational standards and device specifications, thereby underscoring the meticulous orchestration of the overall data processing framework.

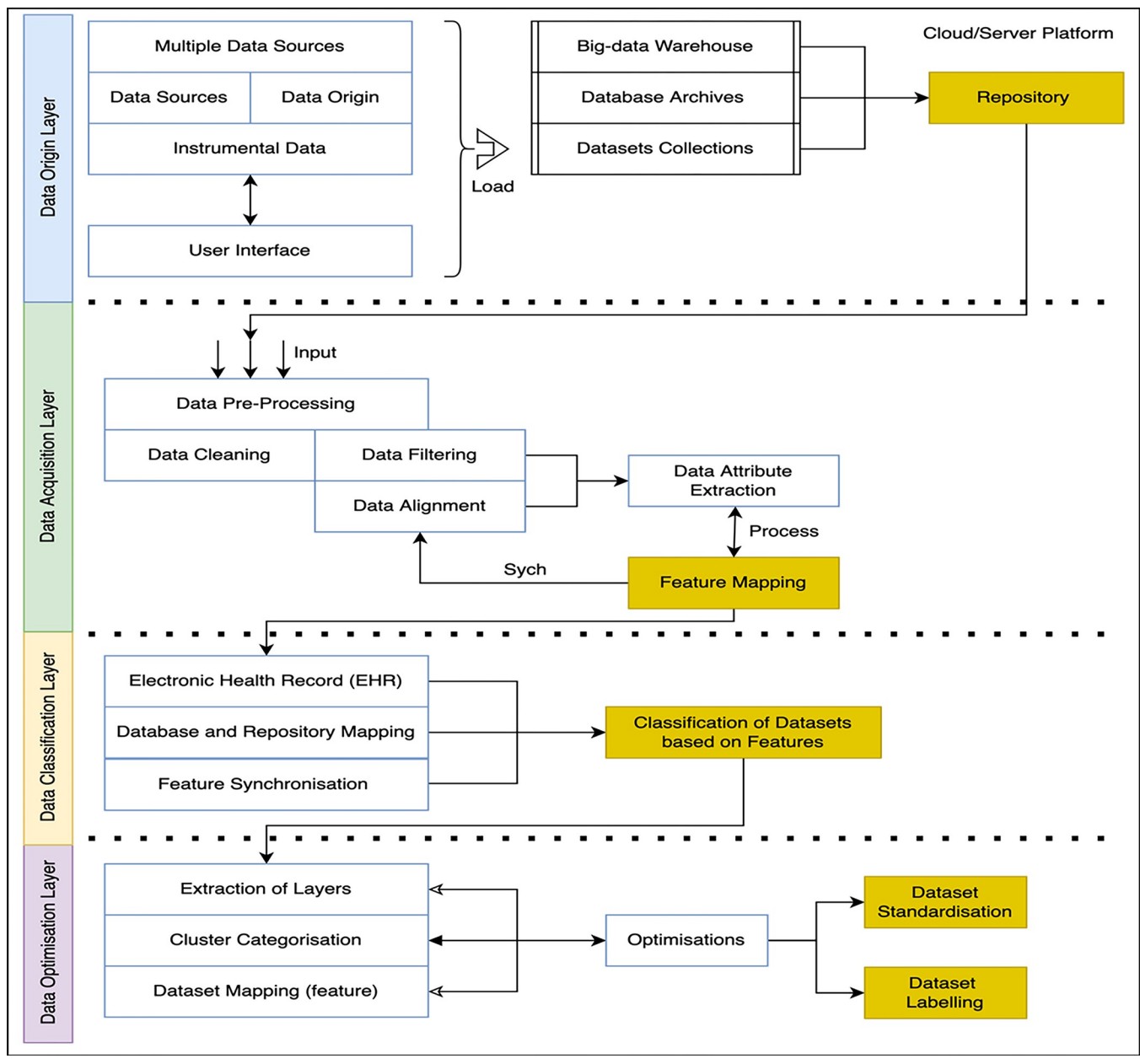

**Fig 1. Block diagram of federated learning model of biomedical data analytics and processing.**

## IV. Mathematical model and hypothesis

The biomedical data is generated via multiple origins sources and servers. These datasets are independent and are dependent based on the scenario and instrumental setup. The dependency is difficult to evaluate since the origin factor of datasets are complex and untracked. Using the proposed technique the framework for optimizing medical datasets in telemedicine is designed and developed. The technique extracts informative attributes set (A) with an origin evaluation parameter (P) such that $(A \Rightarrow \sum p / \forall p \in f)$ where (f) is the feature set associated in attribute mapping (M). Consider the incoming dataset (D) from multiple origin fronts as $D = \{D_1, D_2, D_3 \ldots D_n\}$ where each $(D)_i \Rightarrow \overrightarrow{\Delta A}$ and $(D)_i \Rightarrow \overrightarrow{\Delta O_X}$ where $(\Delta O_X)$ the origin is source

| Edge/Node/Remote device layers | Data Origin Layer | Data Collection \| Source Identification \| Repository Management \| Data Pre-processing \| Filtering \| Warehouse alignment \| Feature-set Mapping |
| --- | --- | --- |
| | Data Acquisition Layer | |
| Data Server Layer | Data Classification Layer | Electronic Health Record Linking \| Classification \| Feature Synchronisation |
| Cloud Layer | Data Optimisation Layer | Data Layers \| Label Mapping \| Data Standardisation \| Optimisation |

**Fig 2. Layers of data processing in Federated Learning (FL) for data streamlining.**

of all computational dataset creation. The origin $O_X$ of source databases is represented as $(\sum_{i\to\infty}(O_X) \Rightarrow \lim_{n\to\infty}(D_i))$ with each origin point reflects the paradigm operation as shown in Eq 1 as $(\omega)$ is the factor of feature association.

$$\lim_{n\to\infty}(O_X) \Rightarrow \left[\frac{\delta(D_X)_i}{\delta t}\right]\sum_{i\in D_X}f_i(\omega) \qquad 1$$

Then the learning sequence of the origin $(O_X)$ is directly dependent on source of resourcing index and path of dataset generation.

$$\lim_{n\to\infty}(O_X) \Rightarrow \sum_{n=i^2}^{\infty}\left\{\left[\frac{\delta(D_X)_i}{\delta t}\right]\left[f_i(\omega) - \overrightarrow{\Delta A}\right]\right\} \qquad 2$$

Where the attribute $(\overrightarrow{\Delta A})$ of each incoming dataset is validated and indexed. The summarization of parameters results in a cumulative database termed as centralized Edge database $(E_X)$, the $(E_X)$ is internally combined with supporting parameters such as $(E_X)^U \Rightarrow (\min_{\omega\in f(i)}(\overrightarrow{\Delta A}))$ with a resulting ecosystem of large $(E_X)$ as shown in Eq 3

$$(E_X) = \lim_{n\to\infty}\left(\beta(\omega_i) \oplus \sum_{i=0}^{n}\left(\frac{\delta(O_X)_i}{\delta t}\right)\right) \qquad 3$$

$$\therefore (E_X) = \lim_{n\to\infty}(\arg\min\beta(\omega_i) \oplus \sum_{i=0}^{n}(\delta(\log_2(O_X)_i))) \qquad 4$$

On customization of source parameters with reference to feature set is as below

$$E_X^{Comp}(f) = [\sum_{i=0}^{\infty}(\beta(\omega_i)|\delta(E_X)_i)] \qquad 5$$

According to simulative agreement the factorial values of dataset origin and other resources need a track and follow-up path accordingly. Considering the path $(\wp)$ with a tracking ratio as

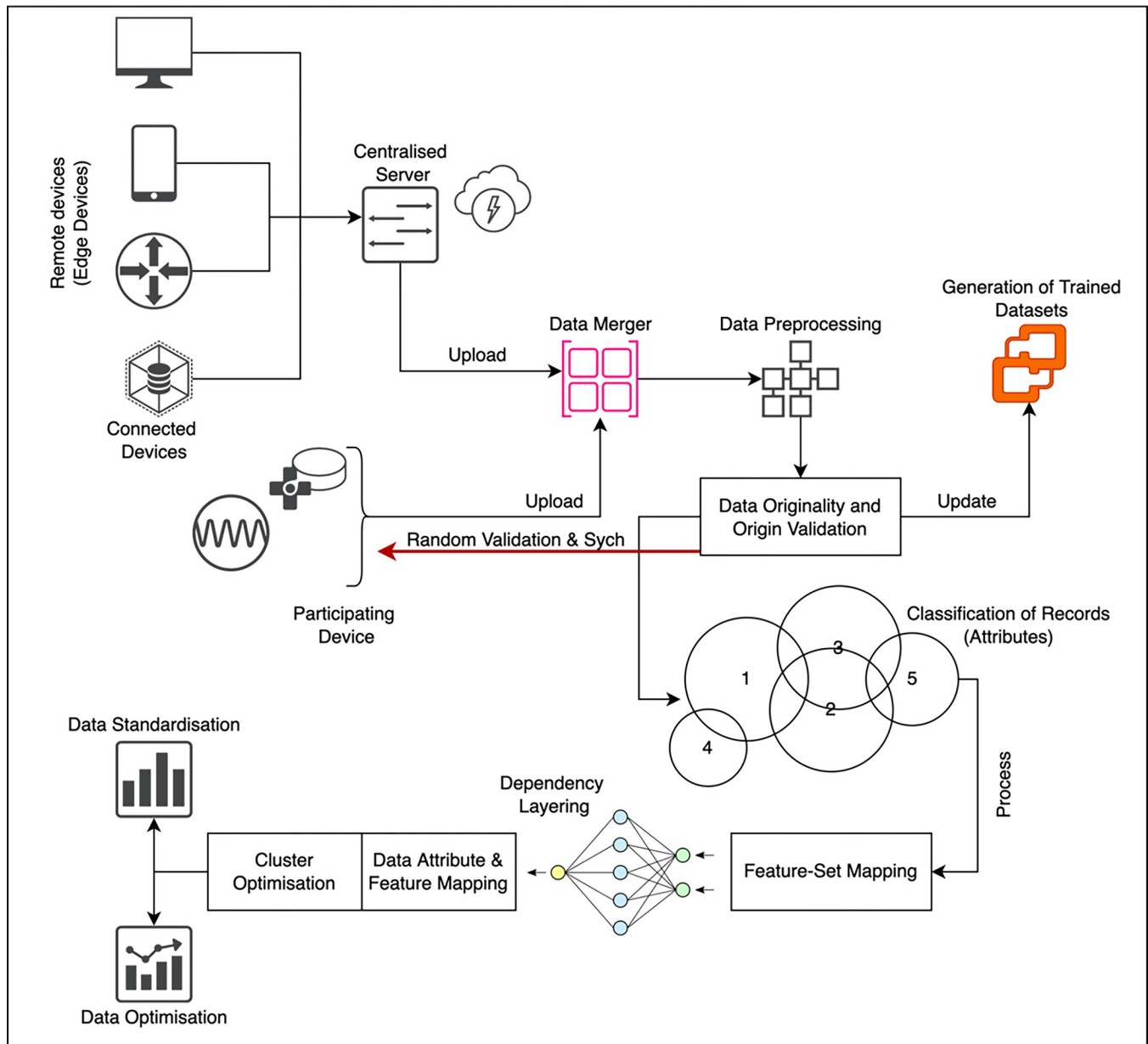

**Fig 3. Architecture diagram of proposed model.**

shown in Eq 6

$$\wp(\xi, \overrightarrow{E_X}) = \left[ \frac{O(\log_2(\beta\omega_i))_0^n}{\Delta t}, \frac{\delta(E_X^{Comp}[f]_i)}{\delta t} \right]$$

The follow-up path ($\wp$) has a with an extraction variable ($\xi$) is evaluated with independent matrix, each matrix is a supporting parameter of previously path-charted values ($\overrightarrow{\wp}$) as $\overrightarrow{\wp} \Rightarrow \wp/\forall\wp(\xi, \overrightarrow{E_X}) \subseteq O_X | O_X \in D$ where ($D$) is the distributed factor of path evaluated.

## Path evaluation and federated learning

Each path $\overrightarrow{\wp}$ is a resultant of multiple data origin $O_X$ and its associated paradigms using the terminology of federated learning, the structural function $f(S)$ is extracted from each path $(\wp)$ as $(\wp \subseteq O_X)$ at instance of each path directed towards the distributed factor $(D)$ based datacenters. Consider the path $(\wp)$ is justified with normal operation strategies. The procedure is dependent on path vector $(\wp_X)$ and routing matrix $(\mathbb{R})$, whereas in federated, the interdependency of each variable associated in paths operation is considered and evaluated as shown in Eq 7

$$\wp\left(\xi, \overrightarrow{E_X}\right) = \left[\frac{\delta(E_X)}{\delta t}\right] \Rightarrow \left[\lim_{n \to \infty}(\Delta\wp_X\{E_X, \Delta\mathbb{R}\})\right] \qquad 7$$

$$\therefore \wp\left(\xi, \overrightarrow{E_X}\right) = \left[\frac{\delta(E_X)}{\delta t}\right]_0^n \Rightarrow \left\{\left[\sum_{i=n}^{\infty}\frac{\delta(\Delta\wp_X\{E_{X_i}, \Delta\mathbb{R}_i\})}{\delta t}\right] \otimes \overrightarrow{\Delta A}\right\} \qquad 8$$

Where $\overrightarrow{\Delta A}$ a fractional value formed by attributes extraction while passing the path and route. Technically, the dependent matrix of each path and route is represented as $M(\wp, \mathbb{R})_X$ such that $(\wp \subseteq O_X \subseteq \mathbb{R})$ and the user $(U)$ in operation is validated with supportive learning. The users that are directly supporting the leaning objective by consent to participate are represented as $(U_C)$ and others under non-consent are $(\bar{U}_C)$ such that, the inter-linear associations of users are extracted to study the pattern of data analysis as shown in Eq 9 with $\Gamma_P$ as evaluation pattern.

$$\Gamma_P = \left[\lim_{n \to \infty}\left(\delta(\xi, \wp_X)_i \oplus \left(\frac{\delta(U_C)}{\delta t} - \frac{\delta(\bar{U}_C)}{\delta t}\right)\right)\right] \qquad 9$$

$$\Gamma_P = \left[\lim_{n \to \infty}\left(\delta(\xi, \wp_X)_i \oplus \sum_{i=0}^{n}\left(\frac{\delta(U_C)_i - \delta(\bar{U}_C)_{i-1}}{\delta t}\right)\right)\right] \qquad 10$$

Where each path $(\wp)$ is dependent on $\Gamma_P$ to assure a reliable matrix in path selection. Thus $\Gamma_P$ extracts the path values from origin source via interdependent features and attributes.

## Federated data-categorization

The extracted data is typically aligned and formatted based on path, series of resource contribution and feature dependency evaluation as $(\wp \subseteq O_X \subseteq \mathbb{R}) \Rightarrow D$ with a path of user participation is finalized. Each dataset from multiple devices are pooled and interdependency features are extracted and represented as $(f_{O_X})$ with a dependency matrix $\wp_X \Rightarrow \mathbb{R}_X \Rightarrow \sum(f_{O_X})$ with each user data is labeled as an individual entity $(f_{O_X})_i$ with $(i \to \infty)$ under the operational principles. Consider the data stream $(D_S)$ stored in $(D)$ is extracted as shown in Eq 11.

$$D_S = Ext[\lim_{n \to \infty}\{\delta(\wp, O_X) \oplus \delta(\mathbb{R}, O_X) \oplus \Gamma_P\}] \qquad 11$$

$$\therefore (D_S)_{min} = Ext\left[\lim_{n \to \infty}\left\{\sum_{i=o}^{n}\sum_{j=i+1}^{n-2}\left(\frac{\delta(\wp_X, \mathbb{R}_X, \Gamma_p)_i}{(O_X)_j}\right)\right\}\right] \qquad 12$$

$$\therefore (D_S)_{min} = \frac{1}{\nabla(O_X)}Ext\left[\int_0^{\infty}\left(\frac{\delta(\wp_X, \mathbb{R}_X, \Gamma_p)_i}{\delta P}\right)\right] \qquad 13$$

Where each $(O_X)_j \Rightarrow (\delta P | \nabla(O_X))$ with a supporting variable matrix to evaluated data indexing

is represented. Typically the evaluation data in $(D_S)_{\min} \Rightarrow \infty(D_S)_0^n$ with a customization database. The orientation of dataset $(D_S)_{\min}$ is a resultant of processed attributes in a medical edge device and origin devices. Thus, the categorization of datasets are typically based on feature-set ($f$) with reference tk origin source ($O_X$) as ($f(x) \Rightarrow O_X$) such that, $\forall (f(x) \Rightarrow (D_S)_{\min})$ with a supporting matrix (D) and threshold feature evaluation time ($\Delta T$) with each associated feature-set.

## Feature-set extraction and evaluation

The process of feature-set extraction is relevance to the applied of feature data categorization. The relevance of information is to assure the processed medical dataset is authenticated and has generated $(O_X)_i$ based origin matrix for evaluation. The feature extraction ($f_e$) is represented from ($\forall f_e \subseteq f$) with a feature dependency matrix to be considered before evaluation. Consider feature selection set ($f_e$) as ($f_e$) = $[f_1^{e^\dagger}, f_2^{e^\dagger}, f_3^{e^\dagger}, f_4^{e^\dagger} \ldots f_n^{e^\dagger}]$ such that ($\forall e \subseteq f_e \in f$) to achieve a feature independent ratio. The feature set of given data (medical) using data-stream ($D_S$) is a bi-product of $(D_S)_{\min} \cup (O_X)_i | (O_X)_j$ under additional feature / attribute mapping as shown in Eq 14.

$$f_e = \Delta(D_S)_0^\infty \,|\, \lim_{n \to \infty} \left( \frac{\delta^2 (D_S)_{\min}}{\delta t^2} \right)_0^{n!} \odot \Delta O_X \tag{14}$$

$$\Rightarrow f_e = \Delta(D_S)_0^\infty \left( 1 - \left\{ \frac{\partial^2 (D_S)_{\min}}{\partial t^2} \otimes \Delta O_X \right\} \right) \tag{15}$$

With instance repeated with multiple occurrences, the rational values of each featureset ($f_e$) is monitored and extracted as shown in Eq 16

$$\therefore \hat{f}_e = \Delta(D_S)_0^\infty \left( 1 - \sum_{i=0}^\infty \sum_{j=n-i}^{n-j^2} \left( \frac{\partial^2 ([D_S]_{\min})_i}{\partial t^2} \oplus (\Delta O_X)_{(i,j)} \right) \right) \tag{16}$$

The feature set extracted from each relevance of origin $\Delta O_X$ is subjected to ($i,j$) such that, $[\forall \Delta O_{X(i,j)} \subseteq \hat{f}_e]$ and has a relatively higher feature dependency matrix. The process of evaluation is further computed using a regional matrix of feature dependencies and reverse origin tracking.

## Federated learning based data optimization and standardization

The major goal of data analytics in biomedical datasets is to achieve a higher reliability factor and accessing for a supportive decision making. Typically, the extracted features from Eq 16 is validated and processed with in a single unit of cloud/server. The terminology of federated learning is to achieve a remote computation using decentralization of data serves and accumulates. The terminology is to support the extracted feature mapping and customization with the other computational devices associated in layer as shown in Fig 2. The variables customization is supported by multiple edge devices $\xi_d$ connected via an agreement portal as ($\forall \xi_d \subseteq \wp | \mathbb{R}$) and each of $\xi_d$ is represented with a follow-up server ($\xi_d)_S$ such that ($\xi_d)_S \subseteq S$ where S is a server of multiple occurrence on S = ($S_1, S_2, S_3 \ldots S_n$). These edge devices are relatively complex on privacy and accessing information. Each edge device ($\xi_d)_S \Rightarrow A(D_S)_{\min}$ is appended to assure the flow of information stream. Typically, the node on each serve can optimize and be feature centric. It additionally monitors the configuration ($C_f$) bound on ($\xi_d)_S$ such that $\forall C_f \in (\xi_d)_S$ at a given interval of time ($t$) as shown in below hypothesis.

 

**Case 1: Optimization of datasets.**

$$(O_D)_{(i,j)} = \circ_{\max} - \left( \lim_{n \to \infty} \left[ \frac{\partial^2(\hat{f}_e)}{\partial t^2} \oplus \frac{\partial^2(\xi_d)_S}{\partial t^2} \right] \right) \qquad 17$$

$$\therefore (O_D)_{(i,j)} = \circ_{\max} - \left( \lim_{n \to \infty} \left[ \frac{\partial^2(\hat{f}_e) \oplus \partial^2(\xi_d)_S}{\partial t^2} \right] \right) \qquad 18$$

On customization of secondary paradigm used by edge devices ($\xi_d$) is reflected and validated as below

$$\therefore (O_D)_{(i,j)} = \circ_{\max} - \left\{ \left( \frac{(O_X)_{\max} \log_2(1/(\xi_d)_S)}{\delta t} \right) \bullet \left( \frac{\partial^2(\hat{f}_e) \oplus \partial^2}{\partial t^2} \right) \right\} \qquad 19$$

On secondary optimizations

$$(O_D)_{(i,j)} = \circ_{\max} - \left\{ \left[ \frac{(O_X)_{\max} \log_2 1/(\xi_d)_S}{\delta t} \oplus \frac{\partial^2(\hat{f}_e)}{\partial t^2} \right] \Delta O_X \right\} \qquad 20$$

The generalization process of each extracted and optimized datasets is filtered and processed with a dual linking and origin parameters. $(\Delta O_X)_{\max}$ and $\Delta O_X$ respectively. These coordinates are responsible to achieve an in-order sequence of extracted feature with respect to origin and path ($\wp$) and route ($\mathbb{R}$) in relevance to decision making. The resulting matrix of information is represented below.

$$(\mathbb{R}_X)_{O_D} = \left[ \frac{(\Gamma_P)_0^\infty - \log_2(\hat{f}_e)}{O_X} \right] \cong \left[ \frac{\partial^2(O_D)_{(i,j)}}{\partial t^2} \right] \qquad 21$$

Where $(\mathbb{R}_X)_{O_D}$ is the representative vertex of multiple parameters extracted via various routers ($\mathbb{R}$), path ($\wp$) and origin ($O_X$) optimized dataset as $\sum (\mathbb{R}_X)_{O_D}$ with each data isolated with operating standards and principles.

**Case 2: Standardization of datasets.** The dataset under consideration with respect to Eq 16 is related to the series of features mapped. In this case, the datasets can be standardized with reference to the operation standards. These standards are typically bound to minimal span operation (MSO) approach. Consider the standardization ($S_D$) with each data dependent of multiple parameters such as attributes ($A$), source of origin ($O_X$) and feature categorization ($f_C$) class. These parameters are further evaluated and reconstructed to attain the standardization process as demonstrated below

$$(S_D) = \sum_{\infty} \left( \arg\min \left( \wp(\xi, \overrightarrow{E_X}) \right) \otimes \frac{\partial^2(\Gamma_P)}{\partial t^2} \right) \Delta O_X \qquad 22$$

Where $\wp(\xi, \overrightarrow{E_X})$ is a communicative path values of extracted datasets to achieve a higher order standardization via multiple occurrences (i.e.) ($\Gamma_P$) on ($\Delta O_X$) respectively with a

relatively mapping of feature set $(\hat{f}_e)$ on indirect indexing, the $S_D$ representation follows as below

$$\Rightarrow S_D = \sum_{i=o}^{\infty} \sum_{j=i}^{n} \left( \arg\min\left( \wp(\xi, \overrightarrow{E_X})_{(i,j)} \right) \times \left( \sum_{k=j_{min}}^{n} \left[ \frac{\partial^2 (\Gamma_P)_k}{\partial t^2} \right] \bullet \Delta O_X \right) \right) \qquad 23$$

$$S_D = \lim_{n \to \infty} \left( \sum_{j,i}^{n} \left( \arg\min\left( \wp(\xi, \overrightarrow{E_X})_{(i,j)} \right) \times \left( \sum_{k=j_{min}}^{n} \left[ \frac{\partial^2 (\Gamma_P)_k}{\partial t^2} \right] \bullet \Delta O_X \right) \right) \right) \qquad 24$$

$$\therefore S_D = \lim_{n \to \infty} \frac{\arg\min(\wp(\xi, \overrightarrow{E_X})_{(i,j)})}{\log_2(O_X)} \otimes \left( \sum_{k=j_{min}}^{n} \left[ \frac{\partial^2 (\Gamma_P)_k}{\partial t^2} \right] \right) \qquad 25$$

On reduction the process of optimized as

$$\therefore S_D = \left\{ \log_2(O_X) \approx \lim_{n \to \infty} \frac{\arg\min(\wp(\xi, \overrightarrow{E_X})_{(i,j)})}{\delta t} \approx \left( \sum_{k=j_{min}}^{n} \left[ \frac{\partial^2 (\Gamma_P)_k}{\partial t^2} \right] \right) \right\} \qquad 26$$

Where each standardization values $(S_D)$ is supportive and functional to the validated path $\wp(\xi, \overrightarrow{E_X})$ with reference to $(\Gamma_P)$ and origin $(O_X)$ such that $\forall (O_X) \subseteq \circ$, where $(\circ)$ is the universal origin standards and operation principles to perform the operations.

## V. Results and discussions

The computational approach of the proposed technique is to assure the data (medical) is under standardization and hence reflects the process of dependency tracking and process evaluation for larger datasets. The technique has compared as paradigm such as Origin of information indexed $(O_X)$, path of uploading $(\wp)$, route sourcing $(\mathbb{R})$, extracted attributes $(\overrightarrow{E_X})$ database and secondary influencing $(\Gamma_P)$ as evaluation parameter to provide a reliable decision support. These paradigms are compared under a regional scaling of data transfer and dependency with reference to data modeling as shown in Table 1.

### Implementation setup

The proposed approach is developed on federated cloud environment on schematic outlines provided by AWS edge servers. The categorization and customization of servers and server configuration is managed using kubernetics (K8s). This is an open source platform for orchestration and scaling of cloud applications. The federated server images are docked under a single-line containers for micro-service creation. The setup is simulated on Ubuntu 20.20 server OS with 128GB RAM and 100GB soft-limit of server space under AWS.

**Table 1. Comparative results and observations of proposed technique.**

| | Cluster Size (C) | | | | | | | | |
|---|---|---|---|---|---|---|---|---|---|
| | **2** | **4** | **8** | **16** | **32** | **64** | **128** | **256** | **542** |
| **Path ($\wp$) (AVGTIM)** | 78.32 | 78.33 | 77.91 | 77.21 | 81.32 | 79.32 | 76.32 | 77.23 | 78.11 |
| **Routing ($\mathbb{R}$)** | 99.32 | 98.32 | 98.01 | 97.11 | 93.21 | 94.21 | 96.221 | 97.11 | 96.92 |
| **Dependency Factor ($D_S$)** | Low | Low | Low | Medium | Medium | Medium | High | High | High |
| **Evaluation Ratio ($\Gamma_P$)** | 92.11 | 89.23 | 88.31 | 85.21 | 84.21 | 88.32 | 84.12 | 86.23 | 88.29 |
| **Accuracy (%)** | 99.32 | 99.11 | 98.31 | 96.32 | 95.39 | 96.11 | 96.43 | 97.32 | 97.34 |

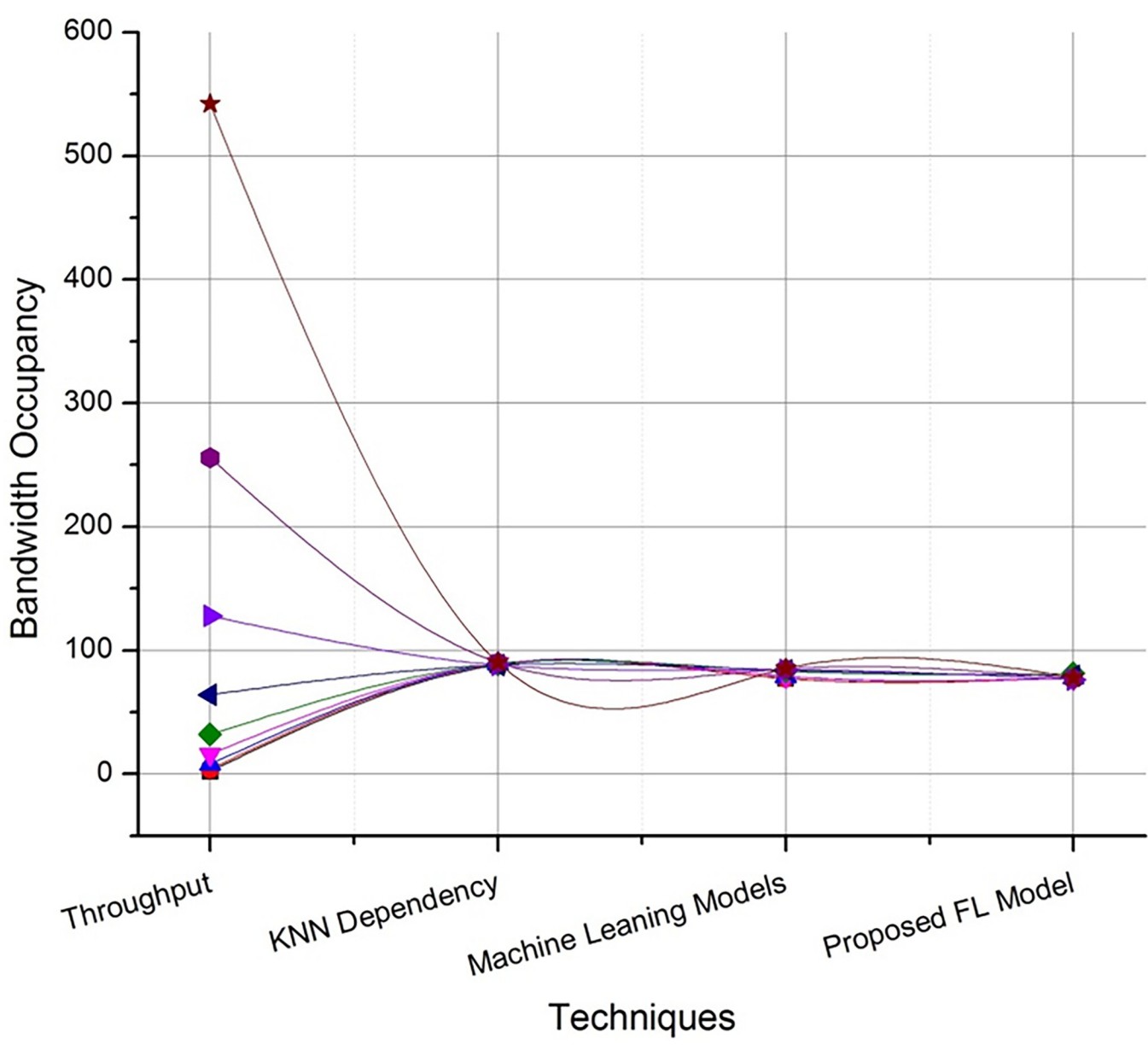

**Fig 4. Comparative evaluation of techniques with federated approach.**

Figs 4 and 5 discusses on the potential representation of comparative analysis and validation. The Fig 4 demonstrates the performance matrix of the proposed technique with across multiple bandwidth. Whereas Fig 5. Deals with the comparison with neural networking (NN) model and machine learning (ML) models with respect to the proposed technique. The proposed Federated (FL) model has outperformed the process with stabilized resource allocation as the cluster size increases. The path parameter is further extended to validate in Figs 6 and 7 respectively based on the routing and evaluation ratio. In general, the proposed federated approach has improved the stabilization ratio of the resources as the cluster size is enhanced.

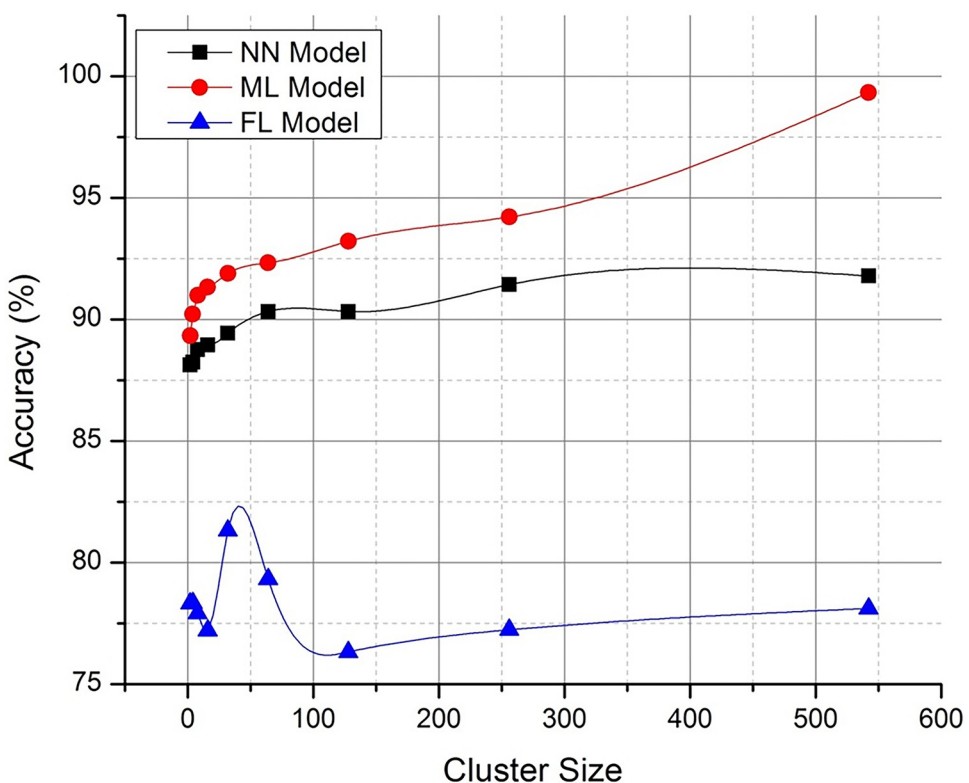

**Fig 5. Path ($\wp$) accuracy tracking (optimized representation).**

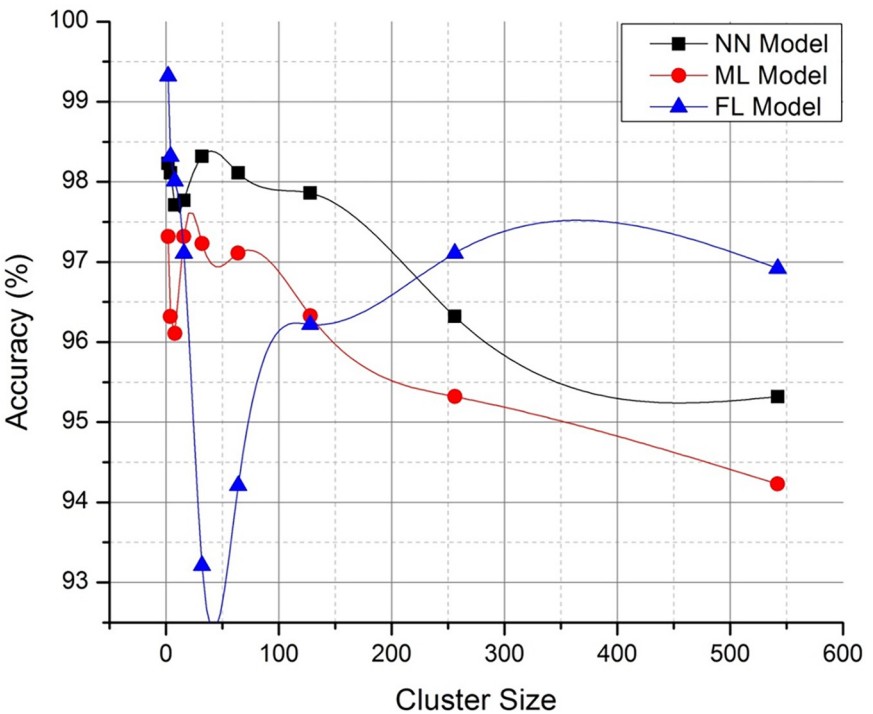

**Fig 6. Routing ($\mathbb{R}$) computation accuracy (optimized representation).**

 

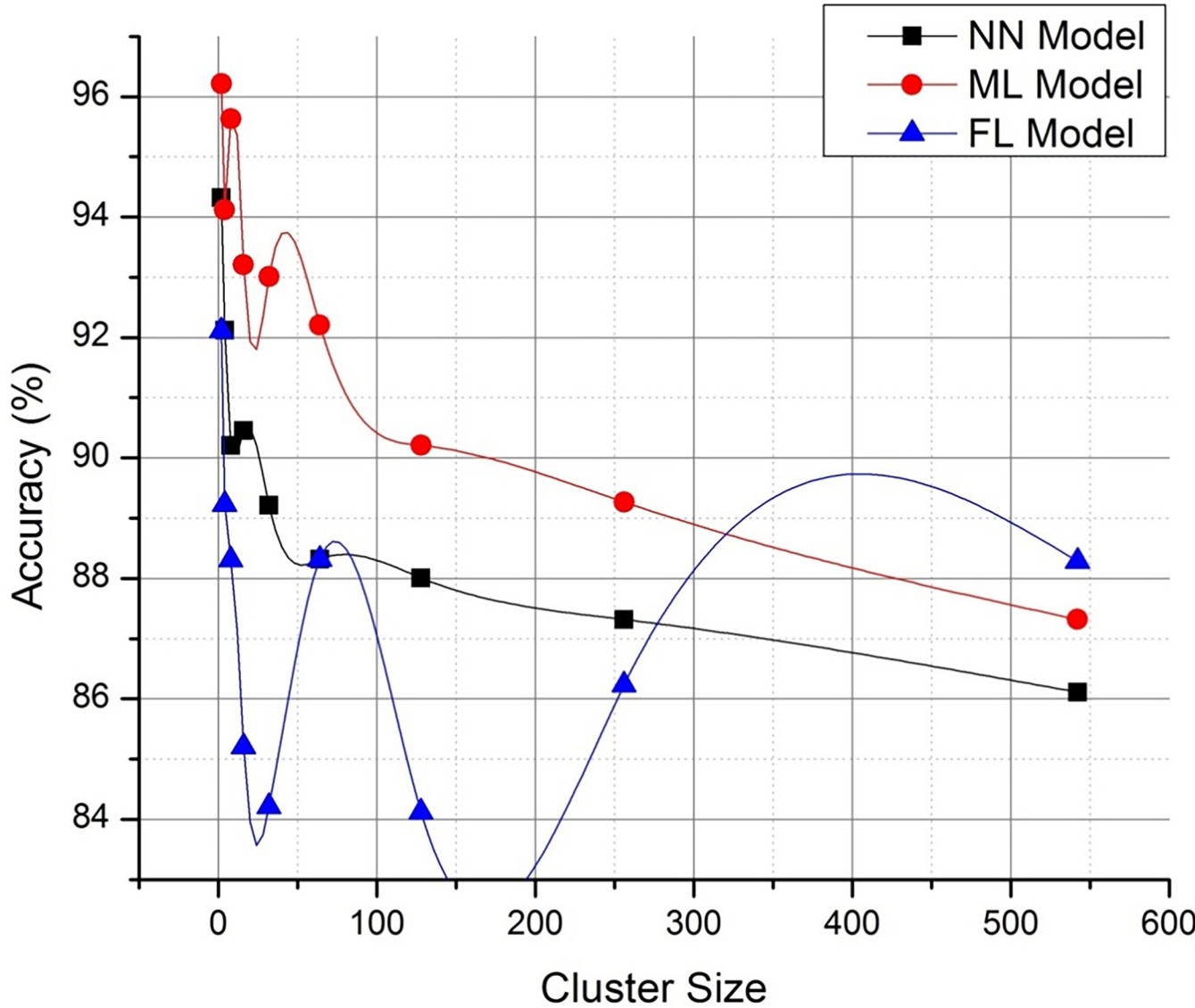

**Fig 7. Evaluation ration ($\Gamma_P$) representation (optimized).**

## VI. Conclusion

The Bigdata analytics of biomedical datasets is a challenging research front. In this paper, a dedicated and novel framework for biomedical data standardization, analysis and optimization is performed using federated learning models. The proposed framework has achieved data optimizations and standardization in three layers of federated modeling as shown in Fig 2. The framework has demonstrated a mathematical proof to interconnect and extract various interdependent features based on paradigm such as Origin of information indexed ($O_X$), path of uploading ($\wp$), route sourcing ($\mathbb{R}$), extracted attributes ($\overrightarrow{E_X}$) database and secondary influencing ($\Gamma_P$) as evaluation parameter to provide a reliable decision support. Overall, the interconnected and aligned processes are aggregated using federated learning approach under data standardization and optimization. The proposed technique has outperformed the existing neural networking models and machine learning models over federated approach. The technique

has demonstrated an accuracy of 97.34% with a cluster size of 542. In near future the technique and framework can be improvised with multiple-layer neural networks based computational models for improved decision making supports.

## Acknowledgments

Z.Z. was partially supported by NIH grant R01LM012806 and the Precision Health Chair Professorship fund. The funders did not participate in the study design, data analysis, decision to publish, or preparation of the manuscript.

## Author Contributions

**Conceptualization:** Afifa Salsabil Fathima.

**Data curation:** Sandeep Kumar Mathivanan, Sukumar Rajendran.

**Formal analysis:** Afifa Salsabil Fathima, Sandeep Kumar Mathivanan.

**Funding acquisition:** Zhongming Zhao.

**Methodology:** Afifa Salsabil Fathima, Syed Muzamil Basha, Syed Thouheed Ahmed.

**Supervision:** Saurav Mallik, Zhongming Zhao.

**Visualization:** Syed Muzamil Basha, Sandeep Kumar Mathivanan.

**Writing – original draft:** Afifa Salsabil Fathima, Saurav Mallik.

**Writing – review & editing:** Saurav Mallik.

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
