## [Decision Letter · Decision Letter 0]

7 Aug 2023

PONE-D-23-21462Federated Learning based Futuristic Biomedical Big-data Analysis and StandardizationPLOS ONE

Dear Dr. M,

Thank you for submitting your manuscript to PLOS ONE. After careful consideration, we feel that it has merit but does not fully meet PLOS ONE’s publication criteria as it currently stands. Therefore, we invite you to submit a revised version of the manuscript that addresses the points raised during the review process.

Based on the reviewer's comments and on my personal observation, I decide that the manuscript can't be accepted in its current form. But authors can go for a major revision as per the given comments. 

Please submit your revised manuscript by Sep 21 2023 11:59PM If you will need more time than this to complete your revisions, please reply to this message or contact the journal office at plosone@plos.org. Please include the following items when submitting your revised manuscript:A rebuttal letter that responds to each point raised by the academic editor and reviewer(s). You should upload this letter as a separate file labeled 'Response to Reviewers'.A marked-up copy of your manuscript that highlights changes made to the original version. You should upload this as a separate file labeled 'Revised Manuscript with Track Changes'.An unmarked version of your revised paper without tracked changes. You should upload this as a separate file labeled 'Manuscript'.

We look forward to receiving your revised manuscript.

Kind regards,

Shitharth Selvarajan

Academic Editor

PLOS ONE

Journal Requirements:

3. Please ensure that you refer to Figures 3,4,5,6 and 7 in your text as, if accepted, production will need this reference to link the reader to the figure.

Additional Editor Comments:

Dear Authors,

Based on the reviewer's comments and on my personal observation, I decide that the manuscript can't be accepted in its current form. But authors can go for a major revision as per the given comments.

Reviewers' comments:

Reviewer's Responses to Questions

**Comments to the Author**

1. Is the manuscript technically sound, and do the data support the conclusions?

Reviewer #1: Yes

Reviewer #2: Yes

2. Has the statistical analysis been performed appropriately and rigorously? 

Reviewer #1: Yes

Reviewer #2: Yes

3. Have the authors made all data underlying the findings in their manuscript fully available?

Reviewer #1: Yes

Reviewer #2: Yes

4. Is the manuscript presented in an intelligible fashion and written in standard English?

Reviewer #1: Yes

Reviewer #2: Yes

5. Review Comments to the Author

Reviewer #1: [1] The abstract should end with a brief statement regarding the significance and impact of this paper.

[2] The method of comparison experiment used in the manuscript is too old. I have not found the recent works published in the year 2022-2023. The following manuscripts can be added.

H. Manoharan, S. Selvarajan, A. Yafoz, H. A. Alterazi, and C. Chen, “Deep Conviction Systems for Biomedical Applications Using Intuiting Procedures With Cross Point Approach,” vol. 10, no. May, pp. 1–14, 2022, doi: 10.3389/fpubh.2022.909628.

S. Selvarajan, H. Manoharan, T. Hasanin, R. Alsini, and M. Uddin, “applied sciences Biomedical Signals for Healthcare Using Hadoop Infrastructure with Artificial Intelligence and Fuzzy Logic Interpretation,” 2022.

H. Manoharan et al., “A machine learning algorithm for classification of mental tasks,” Comput. Electr. Eng., vol. 99, no. February, p. 107785, 2022, doi: 10.1016/j.compeleceng.2022.107785.

P. R. Kshirsagar, H. Manoharan, S. Shitharth, A. M. Alshareef, N. Albishry, and P. K. Balachandran, “Deep Learning Approaches for Prognosis of Automated Skin Disease,” Life, vol. 12, no. 3, p. 426, 2022, doi: 10.3390/life12030426.

A. O. Khadidos, H. Manoharan, S. Selvarajan, and A. O. Khadidos, “A Classy Multifacet Clustering and Fused Optimization Based Classification Methodologies for SCADA Security,” pp. 1–24, 2022.

S. Shitharth, P. Meshram, P. R. Kshirsagar, H. Manoharan, V. Tirth, and V. P. Sundramurthy, “Impact of Big Data Analysis on Nanosensors for Applied Sciences Using Neural Networks,” J. Nanomater., vol. 2021, 2021, doi: 10.1155/2021/4927607.

[3] Abstract should be shortened appropriately. Please reduce the content of the abstract and highlight the merits of the proposed scheme.

[4] The manuscript lacks evaluation indicators, please add several evaluation indicators for further comparative analysis.

[5] The conclusions should explain the comparative results between the proposed and state-of-the-art methods.

[6] The work is well written and well presented but needs to be proofread in English as it has some typos.

[7] Although the authors provide a contextualization of the problem in the introduction, it is not clear what the contribution of the article is.

[8] The authors provided a good description of the works in the literature, but did not provide a detailed description of the difference between the proposed work and other works in the literature. I suggest putting in a table the main characteristics found in the literature and also the proposed work, thus demonstrating the difference between the works.

Reviewer #2: The authors have done a good job on the Federated Learning-based Futuristic Biomedical Big-data Analysis, but this paper still needs improvement.

1. The abstract should always mention the rate of efficacy/efficiency percentage of the proposed method for the reader’s quick overview.

2. The abstract should at least have a line or two about the need for this work. This abstract has an intro and it straightaway deals with the proposed work.

3. Apart from giving the literature in paragraphs, better to add a table which precisely shows the proposed method, pros and cons.

4. The novelty of this paper is not clear. The difference between the present work and previous works should be highlighted. Add more of the issues and what is the significance of this research.

5. The paper lacks a convincing theoretical framework, which is necessary to be considered for publication.

6. Any research paper should include a separate objective section so that readers can easily see the article’s goal. Authors ought to create a separate section for it.

7. Though the paper is mathematically strong, nomenclature should be included. It’s very difficult to follow up on the equations.

8. The use of more parametric comparisons by the authors is recommended. Confusion matrices are a key component for any system’s validity. However, they are seldom mentioned by the authors.

9. Another major issue of this paper is Missing implementation details. The authors haven't mentioned anything about the tool used to build/implement the algorithm. The language they used, and where is the sample code of that? If not at least there should be algorithm’s pseudocode.

10. The references are very less for a scientific article. Among 10 references, not even a single reference is from 2022 and none from 2023. This shows that the paper hasn’t considered any contemporary related works in the survey. I suggest a few more biomedical related papers to cite and refer to enrich the literature.

doi: 10.1109/JSEN.2022.3233407

doi: 10.3389/fpubh.2022.909628

https://doi.org/10.3390/app12105097

11. Authors are advised to follow the IMRAD format for the entire paper.

12. A separate section for Limitations and future work in detail would give further ideas for the readers who wish to enhance your work.

6. PLOS authors have the option to publish the peer review history of their article (what does this mean?). If published, this will include your full peer review and any attached files.

Reviewer #1: No

Reviewer #2: No

---

## [Author Response · Author response to Decision Letter 0]

1 Sep 2023

Reviewer 1: 

[1] The abstract should end with a brief statement regarding the significance and impact of this paper.

Response: We have added the accuracy and correlated performance matrix

[2] The method of comparison experiment used in the manuscript is too old. I have not found the recent works published in the year 2022-2023. The following manuscripts can be added.

• H. Manoharan, S. Selvarajan, A. Yafoz, H. A. Alterazi, and C. Chen, “Deep Conviction Systems for Biomedical Applications Using Intuiting Procedures With Cross Point Approach,” vol. 10, no. May, pp. 1–14, 2022, doi: 10.3389/fpubh.2022.909628.

• S. Selvarajan, H. Manoharan, T. Hasanin, R. Alsini, and M. Uddin, “applied sciences Biomedical Signals for Healthcare Using Hadoop Infrastructure with Artificial Intelligence and Fuzzy Logic Interpretation,” 2022.

• H. Manoharan et al., “A machine learning algorithm for classification of mental tasks,” Comput. Electr. Eng., vol. 99, no. February, p. 107785, 2022, doi: 10.1016/j.compeleceng.2022.107785.

• P. R. Kshirsagar, H. Manoharan, S. Shitharth, A. M. Alshareef, N. Albishry, and P. K. Balachandran, “Deep Learning Approaches for Prognosis of Automated Skin Disease,” Life, vol. 12, no. 3, p. 426, 2022, doi: 10.3390/life12030426.

• A. O. Khadidos, H. Manoharan, S. Selvarajan, and A. O. Khadidos, “A Classy Multifacet Clustering and Fused Optimization Based Classification Methodologies for SCADA Security,” pp. 1–24, 2022.

• S. Shitharth, P. Meshram, P. R. Kshirsagar, H. Manoharan, V. Tirth, and V. P. Sundramurthy, “Impact of Big Data Analysis on Nanosensors for Applied Sciences Using Neural Networks,” J. Nanomater., vol. 2021, 2021, doi: 10.1155/2021/4927607.

Response: We have included the above references.

[3] Abstract should be shortened appropriately. Please reduce the content of the abstract and highlight the merits of the proposed scheme.

Response: Updated

[4] The manuscript lacks evaluation indicators, please add several evaluation indicators for further comparative analysis.

Response: We have updated the comparative analysis models in mathematical section under Table. 1

[5] The conclusions should explain the comparative results between the proposed and state-of-the-art methods.

Response: Performance metric and comparative explanation is added in conclusion.

[6] The work is well written and well-presented but needs to be proofread in English as it has some typos.

Response: English and Typo check is performed 

[7] Although the authors provide a contextualization of the problem in the introduction, it is not clear what the contribution of the article is.

Response: The introduction is rewritten and contribution of article with relevance to the proposed work and filed is highlighted. 

Reviewer 2: The authors have done a good job on the Federated Learning-based Futuristic Biomedical Big-data Analysis, but this paper still needs improvement.

1. The abstract should always mention the rate of efficacy/efficiency percentage of the proposed method for the reader’s quick overview.

Response: We have added the performance metric 

2. The abstract should at least have a line or two about the need for this work. This abstract has an intro and it straightaway deals with the proposed work.

Response: The entire, abstract is rewritten and the comments incorporated. 

3. Apart from giving the literature in paragraphs, better to add a table which precisely shows the proposed method, pros and cons.

Response: The comparative parameters on the study is limited, hence we could not provide a table format of pros and cons

4. The novelty of this paper is not clear. The difference between the present work and previous works should be highlighted. Add more of the issues and what is the significance of this research.

Response: This comment is highlighted in Introduction section

5. The paper lacks a convincing theoretical framework, which is necessary to be considered for publication.

Response: Theoretical explanation is added and enhanced in the section 3

6. Any research paper should include a separate objective section so that readers can easily see the article’s goal. Authors ought to create a separate section for it.

Response: The objective is added in Introduction, the same is highlighted for your kind consideration.

7. Though the paper is mathematically strong, nomenclature should be included. It’s very difficult to follow up on the equations.

Response: We tried our best to optimize the equations in revised 

8. The use of more parametric comparisons by the authors is recommended. Confusion matrices are a key component for any system’s validity. However, they are seldom mentioned by the authors.

Response: Table. 1 discussed on these parameters.

9. Another major issue of this paper is missing implementation details. The authors haven't mentioned anything about the tool used to build/implement the algorithm. The language they used, and where is the sample code of that? If not at least there should be algorithm’s pseudocode.

Response: The implementation details are added in results and discussion section. 

10. The references are very less for a scientific article. Among 10 references, not even a single reference is from 2022 and none from 2023. 

Response: We have updated our reference section

11. Authors are advised to follow the IMRAD format for the entire paper.

Response: We have assured the paper is in IMRAD format 

12. A separate section for Limitations and future work in detail would give further ideas for the readers who wish to enhance your work.

Response: We have considered to keep it in conclusion.

1. We notice that your manuscript file was uploaded on July 10, 2023. Please can you upload the latest version of your revised manuscript as the main article file, ensuring that does not contain any tracked changes or highlighting. This will be used in the production process if your manuscript is accepted. Please follow this link for more information: http://blogs.PLOS.org/everyone/2011/05/10/how-to-submit-your-revised-manuscript/

We have not made the manuscript without any highlights or tracking

2. Please upload a Response to Reviewers letter which should include a point by point response to each of the points made by the Editor and / or Reviewers. (This should be uploaded as a 'Response to Reviewers' file type.) Please follow this link for more information: http://blogs.PLOS.org/everyone/2011/05/10/how-to-submit-your-revised-manuscript/

Yes attached.

3. Please ensure that you refer to Figures 3,4,5,6 and 7 in your text as, if accepted, production will need this reference to link the reader to the figure.

Yes, we have cited all the Figures, 3 4 5 6 and 7

4. Please ensure that you refer to Table 1 in your text as, if accepted, production will need this reference to link the reader to the Table.

Yes, we have cited the Table. 1 is the text

---

## [Editor Report · Decision Letter 1]

4 Sep 2023

Federated Learning based Futuristic Biomedical Big-data Analysis and Standardization

PONE-D-23-21462R1

Dear Authors,

We’re pleased to inform you that your manuscript has been judged scientifically suitable for publication and will be formally accepted for publication once it meets all outstanding technical requirements.

Kind regards,

Shitharth Selvarajan

Academic Editor

PLOS ONE

Additional Editor Comments (optional):

The revised version is satisfying and hence it is be accepted in it's current form.
---

## [Editor Report · Acceptance letter]

25 Sep 2023

PONE-D-23-21462R1 

Federated Learning based Futuristic Biomedical Big-data Analysis and Standardization 

Dear Dr. Mathivanan:

I'm pleased to inform you that your manuscript has been deemed suitable for publication in PLOS ONE. Congratulations! Your manuscript is now with our production department. 

Kind regards, 

on behalf of

Dr. Shitharth Selvarajan 

Academic Editor

PLOS ONE